# In situ microbeam surface X-ray scattering reveals alternating step kinetics during crystal growth

Guangxu Ju [1,6✉], Dongwei Xu [1,2], Carol Thompson [3], Matthew J. Highland[4], Jeffrey A. Eastman [1], Weronika Walkosz[5], Peter Zapol [1] & G. Brian Stephenson[1✉]

The stacking sequence of hexagonal close-packed and related crystals typically results in steps on vicinal {0001} surfaces that have alternating $A$ and $B$ structures with different growth kinetics. However, because it is difficult to experimentally identify which step has the $A$ or $B$ structure, it has not been possible to determine which has faster adatom attachment kinetics. Here we show that in situ microbeam surface X-ray scattering can determine whether $A$ or $B$ steps have faster kinetics under specific growth conditions. We demonstrate this for organo-metallic vapor phase epitaxy of (0001) GaN. X-ray measurements performed during growth find that the average width of terraces above $A$ steps increases with growth rate, indicating that attachment rate constants are higher for $A$ steps, in contrast to most predictions. Our results have direct implications for understanding the atomic-scale mechanisms of GaN growth and can be applied to a wide variety of related crystals.

[1] Materials Science Division, Argonne National Laboratory, Lemont, IL, USA. [2] School of Energy and Power Engineering, Huazhong University of Science and Technology, Wuhan, Hubei, China. [3] Department of Physics, Northern Illinois University, DeKalb, IL, USA. [4] X-ray Science Division, Argonne National Laboratory, Lemont, IL, USA. [5] Department of Physics, Lake Forest College, Lake Forest, IL, USA. [6] Present address: Lumileds Lighting Co., San Jose, CA, USA. ✉email: juguangxu@gmail.com; stephenson@anl.gov

Our understanding of crystal growth is built on a powerful paradigm quantified by Burton, Cabrera, and Frank (BCF)[1-3], in which atoms are added to the growing crystal surface by attachment at the steps forming the edges of each exposed atomic layer, or terrace. The BCF model was originally developed for crystals with step heights of a full unit-cell and step properties that are identical from step to step for a given step direction. However, from the beginning[4] it was recognized that there could be more complex situations. When the space group of the crystal includes screw axes or glide planes, the growth behavior can be fundamentally different on facets perpendicular to one of these symmetry elements[5]. In this case, the terraces can still all have the same atomic arrangement, but now have different in-plane orientations of their top layer. The fractional-unit-cell-height steps that separate these terraces have structures and properties that can vary from step to step, even for a fixed step direction. Thus, surface morphologies with alternating terrace widths can arise that depend upon the deposition or evaporation conditions, as indicated in Fig. 1. The inequivalent kinetics at steps affects not only surface morphology but also the incorporation of alloying elements during crystal growth[6,7].

A ubiquitous but subtle version of this effect occurs on the basal-plane {0001}-type surfaces of crystals having hexagonal close-packed (HCP) or related structures, which are normal to a $6_3$ screw axis. Such crystals are made up of closely packed layers with 3-fold symmetry that alternate between opposite orientations, as shown by the $\alpha$ and $\beta$ terrace structures in Fig. 2c. On a vicinal surface, the $\alpha\beta\alpha\beta$ stacking sequence typically results in half-unit-cell-height steps. The lowest energy steps are normal to $\langle 01\bar{1}0 \rangle$-type directions, and have alternating structures conventionally labeled $A$ and $B$[8,9] as shown in Fig. 2a, b. When the in-plane azimuth of a step changes by 60°, for example, from $[01\bar{1}0]$ to $[10\bar{1}0]$, its structure changes from $A$ to $B$ or $B$ to $A$.

The alternating nature of the steps on such surfaces has been imaged in several systems, including SiC[4], GaN[6,8,10-13], AlN[14], and ZnO[15]. These systems typically show a tendency for local pairing of steps (i.e., alternating step spacings), and an interlaced structure in which the step pairs switch partners at corners where their azimuth changes by 60°, as shown in Fig. 2d. These features are consistent with predictions that $A$ and $B$ steps have significantly different attachment kinetics[6,8,12,16-22] that lead to unequal local fractions of $\alpha$ and $\beta$ terraces during growth.

However, it has not been possible to experimentally distinguish the terrace orientation or step structure, and thus to determine whether $A$ or $B$ steps have faster kinetics.

In particular, the properties of $A$ and $B$ steps on GaN (0001) surfaces have been a matter of some disagreement. A seminal study[8] of molecular beam epitaxy (MBE) of GaN observed alternating step shapes and proposed that the kinetic coefficients for adatom attachment are higher for $A$ steps than $B$ steps, that is, $A$ steps grow faster for a given supersaturation. The support for this highly cited prediction is based on an argument regarding the difference in dangling bonds between $A$ and $B$ steps, and a comparison with experimental results on GaAs (111)[23,24]. (Such face-centered cubic materials have $A$- and $B$-type steps that do not alternate between successive terraces and thus can be distinguished by their orientation[9].) In contrast, several subsequent theoretical studies of GaN (0001) organo-metallic vapor phase epitaxy (OMVPE) and MBE have consistently predicted that $A$ steps have smaller adatom attachment coefficients than $B$ steps. Kinetic Monte Carlo (KMC) studies of GaN (0001) growth under OMVPE conditions found step pairing[17] driven by faster kinetics at $B$ steps than $A$ steps[16]. The standard bond-counting energetics used in a KMC study of growth on an HCP lattice[19] result in a much lower Ehrlich–Schwoebel (ES) barrier at $B$ steps than at $A$ steps, when only nearest-neighbor jumps are allowed. A recent KMC study of GaN (0001) growth under MBE conditions[18] found triangular islands that close analysis reveals are bounded by $A$ steps, indicating faster growth of $B$ steps. An analysis of InGaN (0001) growth by MBE[6] concluded that adatom attachment at $B$ steps is faster, converting them into crenelated edges terminated by $A$ steps.

The difference between the kinetics at $A$ and $B$ steps is a reflection of the chemical states of the adatoms, steps, and terraces that affect the dynamics of $A$ and $B$ steps. Studies of islands on the FCC Pt (111) surface[25,26] have found that $A$ steps have a higher growth rate than $B$ steps, but that this relationship is reversed by the presence of adsorbates such as CO. An experimental study of AlN (0001) surfaces grown by OMVPE[14] found a change in the terrace fraction as a function of the V/III ratio used during growth. Ab initio calculations of kinetic barriers on GaN and AlN (0001) under MBE and OMVPE conditions[20-22] found that the barriers and adsorption energies at $A$ and $B$ steps depend

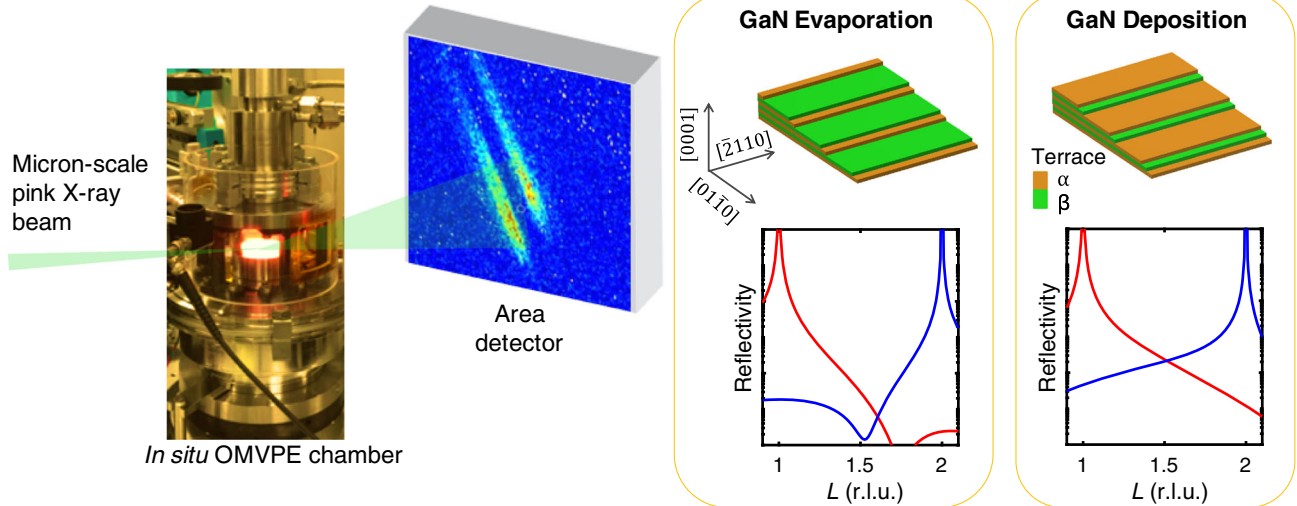

**Fig. 1 Schematic of microbeam surface X-ray scattering during organo-metallic vapor phase epitaxy (OMVPE) growth.** Crystal truncation rod (CTR) measurements are sensitive to the $\alpha$ or $\beta$ terrace fraction changes that occur on vicinal {0001} surfaces of HCP-type crystals during deposition or evaporation. Calculated reflectivities are shown for the CTRs from the (01$\bar{1}$1) and (01$\bar{1}$2) Bragg peaks (red and blue curves, respectively) with $\alpha$ terrace fractions $f_\alpha = 0.1$ and 0.9 typical for evaporation and deposition of GaN by OMVPE, as will be shown below.

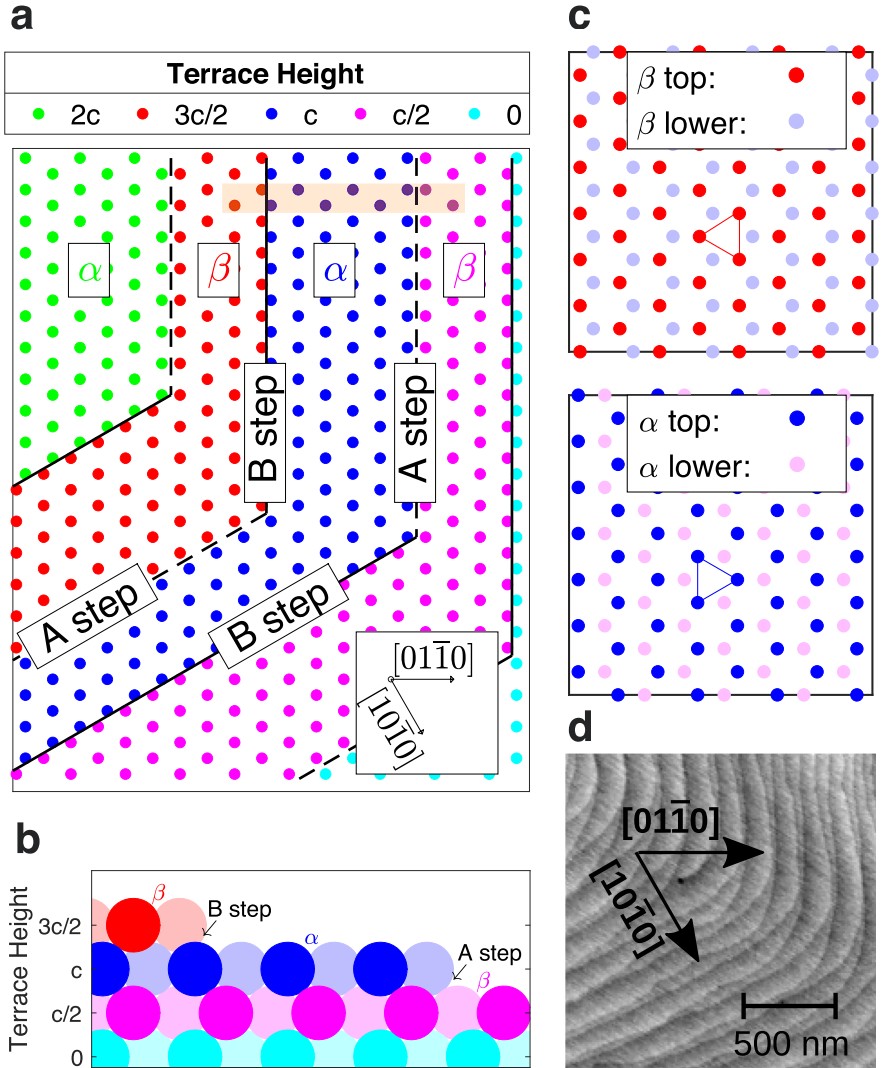

**Fig. 2 Terrace and step structure of vicinal (0001) surface of an HCP-type crystal.** For GaN, only Ga atoms are shown. **a** Circles show top-layer sites on each terrace, with color indicating height. Steps typically have lowest edge energy when they are normal to $[01\bar{1}0]$, $[10\bar{1}0]$, or $[1\bar{1}00]$. Steps in a sequence have alternating structures, $A$ and $B$, which swap when step azimuth changes by 60°. **b** Cross-section of the $A$ and $B$ step structures in the region marked by an orange rectangle in **a**. Lighter and darker colors indicate atoms in different rows. **c** Detail of $\alpha$ and $\beta$ terrace structures. Orientation of triangle of top-layer atoms around $6_3$ screw axis shows difference between layers. **d** AFM height image of GaN (0001) surface typical of films grown on sapphire substrates by OMVPE, showing regions of alternating step spacings and interlacing at corners where the step azimuth changes. Step heights are $c/2 = 2.6$ Å.

in detail on the surface reconstruction induced by the environment. Thus, to properly understand, model, and control (0001) surface morphology in HCP-type systems, there is a need for an in situ experimental method that can distinguish adatom attachment kinetics at $A$ and $B$ steps in the relevant growth environment.

Here, we show that in situ surface X-ray scattering can distinguish the fraction of the surface covered by $\alpha$ or $\beta$ terraces during growth, unambiguously determining differences in the attachment kinetics at $A$ and $B$ steps. X-rays are an ideal probe since they are sensitive to atomic-scale structure and can penetrate the growth environment. This method is enabled by using a micron-scale X-ray beam that illuminates a surface region of a high-quality single crystal having a uniform step azimuth, as shown in Fig. 1. We demonstrate this for OMVPE of (0001) GaN, with measurements of crystal truncation rods (CTRs) carried out in situ during growth. CTRs are streaks of intensity extending in reciprocal space away from every Bragg peak in the direction

normal to the crystal surface, which are sensitive to the surface structure[27]. We fit calculated CTRs from a model structure to these measurements to obtain the variation of the steady state $\alpha$ terrace fraction $f_\alpha$ as a function of growth conditions, as well as the relaxation times $t_{rel}$ of $f_\alpha$ upon changing conditions. These results are compared to calculated dynamics based on a BCF model for a system with alternating step types to quantify the differences in the attachment rates at $A$ and $B$ steps.

## Results

**Calculated surface X-ray scattering with alternating step types.** We present calculations showing how the intensity distribution along the CTRs is sensitive to the fraction of the surface covered by $\alpha$ or $\beta$ terraces. Our calculations include the effect of surface reconstruction, using relaxed atomic coordinates that have been obtained previously[28]. For a vicinal surface, the CTRs are tilted away from the crystal axes, so that the CTRs from different Bragg

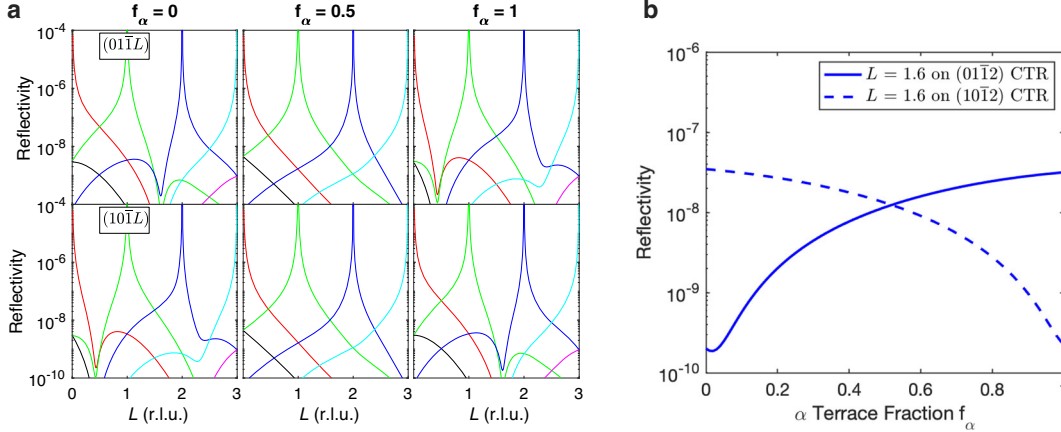

**Fig. 3 Calculated CTR intensities for a vicinal GaN (0001) surface. a** Top and bottom rows show the families of CTRs along $(01\bar{1}L)$ and $(10\bar{1}L)$, respectively. Black, red, green, blue, cyan, and magenta curves are for CTRs from the $L_0 = -1$ to 4 Bragg peaks, respectively. Values of $f_\alpha$ for each column are given at the top. **b** Reflectivity of selected CTRs as a function of terrace fraction $f_\alpha$ for fixed values of $L = 1.6$.

| Table 1 Growth conditions studied. | | | | | | |
|---|---|---|---|---|---|---|
| Growth condition index | TEGa flow ($\mu$mol min$^{-1}$) | H$_2$ frac. in carrier (%) | Net growth rate G (monolayer s$^{-1}$) | | Measured values from CTR fits | Best fit from BCF model |
| 1 | 0.000 | 50 | −0.0018 | $f_\alpha^{ss}$ | 0.111 ± 0.013 | 0.136 |
| 2 | 0.000 | 0 | 0.0000 | $f_\alpha^{ss}$ | 0.461 ± 0.018 | 0.440 |
| 3 | 0.033 | 50 | 0.0109 | $f_\alpha^{ss}$ | 0.811 ± 0.014 | 0.836 |
| 4 | 0.033 | 0 | 0.0127 | $f_\alpha^{ss}$ | 0.868 ± 0.011 | 0.847 |
| 1–2 | | | | $t_{rel}$ | 2200 ± 200 s | 2478 |
| 2–4 | | | | $t_{rel}$ | 340 ± 30 s | 331 |

Also given are values of steady state terrace fraction $f_\alpha^{ss}$ obtained from fits to measured CTR intensities (shown in Fig. 5), values of relaxation time $t_{rel}$ of $f_\alpha$ upon changing conditions (shown in Fig. 6), and corresponding values from the BCF model fit (shown in Fig. 8).

peaks do not overlap. The X-ray reflectivity along the CTRs can be calculated by adding the complex amplitudes from the substrate crystal and the reconstructed overlayers, with proper phase relationships[29–31]. Details of our calculations are given in a separate paper[32].

Figure 3a shows calculated intensity distributions along $(01\bar{1}L)$ and $(10\bar{1}L)$ CTRs for the GaN (0001) surface, demonstrating how their $L$-dependences vary with $f_\alpha$. Bragg peak locations have integer indices $H_0K_0L_0$ in reciprocal lattice units; these indices identify the CTR associated with each peak. For this comparison, we use the 3H(T1) surface reconstruction[28] and a fixed surface roughness, independent of $f_\alpha$, as discussed in the "Methods" section below and Supplementary Discussion 1. The same qualitative behavior is obtained using other surface reconstructions. For $f_\alpha = 0$ and $f_\alpha = 1$, the regions between the Bragg peaks have alternating stronger and weaker intensities, with the alternation being opposite for $(01\bar{1}L)$ and $(10\bar{1}L)$. For $f_\alpha = 0.5$, the intensities between all Bragg peaks are about the same, and there is no difference between the $(01\bar{1}L)$ and $(10\bar{1}L)$ CTRs. As required by symmetry, the $(01\bar{1}L)$ CTRs with $f_\alpha = X$ are identical to the $(10\bar{1}L)$ CTRs with $f_\alpha = 1 - X$, for any value $X$. Figure 3b shows calculations of the reflectivity as a function of $f_\alpha$ at positions near $L = 1.6$ on the $(01\bar{1}2)$ and $(10\bar{1}2)$ CTRs. The variation in reflectivity is almost monotonic in $f_\alpha$ at these positions. These curves are used below to extract $f_\alpha(t)$ during dynamic transitions.

**In situ X-ray scattering measurements during growth.** We studied four OMVPE conditions having different net growth rates at the same temperature 1076 ± 5 K, summarized in Table 1 (see

"Methods" for further details). The substrate used was a GaN single crystal. Figure 4a shows its initial surface morphology determined by ex situ atomic force microscopy (AFM). One can see straight steps almost perpendicular to the $y$ or $[01\bar{1}0]$ direction over large areas. An analysis of the step spacing shows a slight tendency towards pairing, with one of the two alternating terrace types having an area fraction of 0.47. AFM is insensitive to whether this fraction corresponds to the $\alpha$ or $\beta$ terraces. We also characterized the miscut angle by measuring the splitting of the CTRs. Figure 4b shows a transverse cut through the CTRs in the $Q_y$ direction near $(000L)$ at $L = 0.9$. Both the AFM and X-ray measurements give a double-step spacing of $w = 573$ Å corresponding to a miscut angle of 0.52°. To relate the $\alpha$ terrace fraction to the behavior of $A$ and $B$ steps, it is critical to determine the sign of the step azimuth. By making measurements as a function of $L$, we verified that the peak at high $Q_y$ is the CTR coming from (0000), while the peak at low $Q_y$ is the (0002) CTR. This confirms that the downstairs direction of the vicinal surface is in the $+y$ direction, as drawn in Fig. 1. It is also useful to know the precise angle of the step azimuth with respect to the crystal planes, which determines the minimum kink density and thus the predicted values of some kinetic coefficients. X-ray measurements found this to be 5° off of the $[01\bar{1}0]$ direction towards $[10\bar{1}0]$, which gives a maximum kink spacing of 33 Å. The kink spacing could be smaller due to thermally generated kinks[1], as discussed in Supplementary Discussion 2. With this low-dislocation-density substrate and the low growth rates used, the previously reported instability to step bunching during growth[33] was not observed.

Figure 5 shows the measured steady state CTR intensities as a function of $L$, for both the $(01\bar{1}L)$ and $(10\bar{1}L)$ CTRs and at all four conditions. The qualitative behavior agrees with that

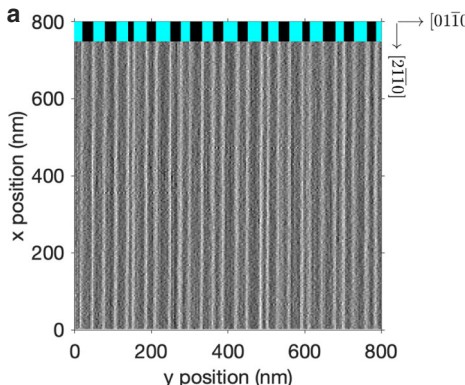
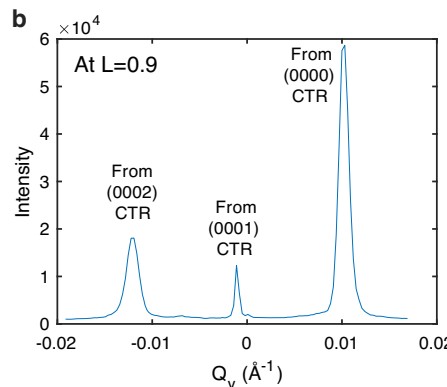

**Fig. 4 Imaging and scattering from steps. a** AFM image of steps of height $c/2$ (2.6 Å) on the vicinal GaN substrate used in X-ray measurements. To emphasize the positions of steps, we plot the amplitude error signal, which is proportional to the height gradient in the scan direction ($y$). Image was obtained ex situ at room $T$ after an anneal for 300 s at 1118 K in zero-growth conditions (0% $H_2$, 0 TEGa). Average fraction over a $2 \times 2\,\mu m^2$ area of terrace family marked black at top is 0.47. The average double-step spacing of $w = 573$ Å corresponds to a miscut angle of $\tan^{-1}(c/w) = 0.52°$. **b** Profile of split CTRs from the vicinal surface, measured at $T = 1170$ K during growth at 0.053 $\mu$mol min$^{-1}$ TEGa and 50% $H_2$. The splitting of the CTRs $\Delta Q_y = 0.0110$ Å$^{-1}$ corresponds to a miscut angle of $\tan^{-1}[\Delta Q_y/(2\pi/c)] = 0.52°$, in agreement with the AFM result.

expected from a variation in $f_\alpha$ shown in Fig. 3a, with alternating higher and lower intensities between the Bragg peaks under some conditions, and opposite behavior of the two CTRs.

**Steady state and dynamics of terrace fraction during growth.** To obtain values of the steady state terrace fraction $f_\alpha^{ss}$ for each of the four conditions, we fit calculated CTR intensities to the measured profiles. We performed fits using different possible surface reconstructions (see "Methods" for further details). While the 3H(T1) reconstruction gives the best fit to all conditions, similar $f_\alpha^{ss}$ values are obtained using alternative reconstructions. For each condition, both the $(01\bar{1}L)$ and $(10\bar{1}L)$ CTRs were simultaneously fit. The values of $f_\alpha^{ss}$ obtained as a function of net growth rate $G$ are given in Table 1. The marked increase in $f_\alpha^{ss}$ as $G$ is increased reveals the qualitative difference between the kinetics at $A$ and $B$ steps during OMVPE of GaN: adatom attachment coefficients for $A$ steps are larger. Thus, a surface with initially balanced $\alpha$ and $\beta$ terrace fractions at zero net growth rate will evolve to one with higher $f_\alpha^{ss}$ during positive net growth, because of the initially higher adatom attachment rate at the $A$ steps. Likewise, during evaporation the initially higher detachment rate at $A$ steps will give a lower $f_\alpha^{ss}$.

We also observed the dynamics of the change in $f_\alpha$ by recording the intensity at a fixed detector position as a function of time before and after an abrupt change between conditions, as shown in Fig. 6a. We chose positions near $L = 1.6$ where the X-ray reflectivity $R$ changes almost monotonically with $f_\alpha$, as shown in Fig. 3b. It is thus straightforward to obtain $f_\alpha(t)$ from the intensity evolution using the calculated $R(f_\alpha)$, as shown in Fig. 6b. The characteristic $1/e$ relaxation times $t_{rel}$ were $2200 \pm 200$ and $340 \pm 30$ s for the transitions from conditions 1 to 2 and 2 to 4, respectively.

**BCF model for surface with alternating step types.** To quantitatively relate the behavior of the terrace fraction to the kinetic properties of $A$ and $B$ steps, we have developed a model[34] based on BCF theory. Such models have been used extensively to understand growth behavior such as the step-bunching instability[35], pairing of steps[36], and competitive adsorption[37], typically where all steps in a sequence are equivalent. In our model, we consider an alternating sequence of two types of terraces, $\alpha$ and $\beta$, and two types of steps, $A$ and $B$, with properties that can differ, as shown in Fig. 7. Related BCF models with an alternating step or terrace properties

have appeared previously[12,16,17,38–40]. We include the effects of step transparency[41] (i.e., adatom transmission across steps) and step–step repulsion[2].

The rate of change in the adatom density per unit area $\rho_i$ on terrace type $i = \alpha$ or $\beta$ is written as

$$\frac{\partial \rho_i}{\partial t} = D\nabla^2 \rho_i - \frac{\rho_i}{\tau} + F, \qquad (1)$$

where $D$ is the adatom diffusivity, $\tau$ is the adatom lifetime before evaporation, and $F$ is the deposition flux of adatoms per unit time and area. The four boundary conditions for the flux at the steps terminating each type of terrace can be written as

$$-D\nabla \rho_\alpha^+ = +\kappa_-^A(\rho_\alpha^+ - \rho_{eq}^A) + \kappa_0^A(\rho_\alpha^+ - \rho_\beta^-), \qquad (2)$$

$$-D\nabla \rho_\alpha^- = -\kappa_+^B(\rho_\alpha^- - \rho_{eq}^B) - \kappa_0^B(\rho_\alpha^- - \rho_\beta^+), \qquad (3)$$

$$-D\nabla \rho_\beta^+ = +\kappa_-^B(\rho_\beta^+ - \rho_{eq}^B) + \kappa_0^B(\rho_\beta^+ - \rho_\alpha^-), \qquad (4)$$

$$-D\nabla \rho_\beta^- = -\kappa_+^A(\rho_\beta^- - \rho_{eq}^A) - \kappa_0^A(\rho_\beta^- - \rho_\alpha^+). \qquad (5)$$

As shown in Fig. 7, $\kappa_+^j$ and $\kappa_-^j$ are the kinetic coefficients for adatom attachment at a step of type $j = A$ or $B$ from below or above, respectively, and $\kappa_0^j$ is the kinetic coefficient for transmission across the step. The $+$ or $-$ superscripts on $\rho_i$ and $\nabla \rho_i$ indicate evaluation at the downhill or uphill terrace boundaries, respectively. We consider the overall vicinal angle of the surface to fix the sum $w$ of the widths of $\alpha$ and $\beta$ terraces, which are thus $f_\alpha w$ and $(1 - f_\alpha)w$. We also assume relations between the equilibrium adatom densities $\rho_{eq}^j$ at the steps and the terrace widths that reflect an effective repulsion between the steps owing to entropic and strain effects[2],

$$\rho_{eq}^j = \rho_{eq}^0 \exp(\mu_j/kT), \qquad (6)$$

where $\rho_{eq}^0$ is the equilibrium adatom density at zero growth rate, and the adatom chemical potentials $\mu_j$ at steps of type $j = A$ or $B$ have a dependence on $f_\alpha$ given by

$$\frac{\mu_A}{kT} = -\frac{\mu_B}{kT} = M(f_\alpha) \equiv \left(\frac{\ell}{w}\right)^3 \left[\left(\frac{1-f_\alpha^0}{1-f_\alpha}\right)^3 - \left(\frac{f_\alpha^0}{f_\alpha}\right)^3\right], \qquad (7)$$

where $\ell$ is the step repulsion length and $f_\alpha^0$ is the terrace fraction at zero growth rate.

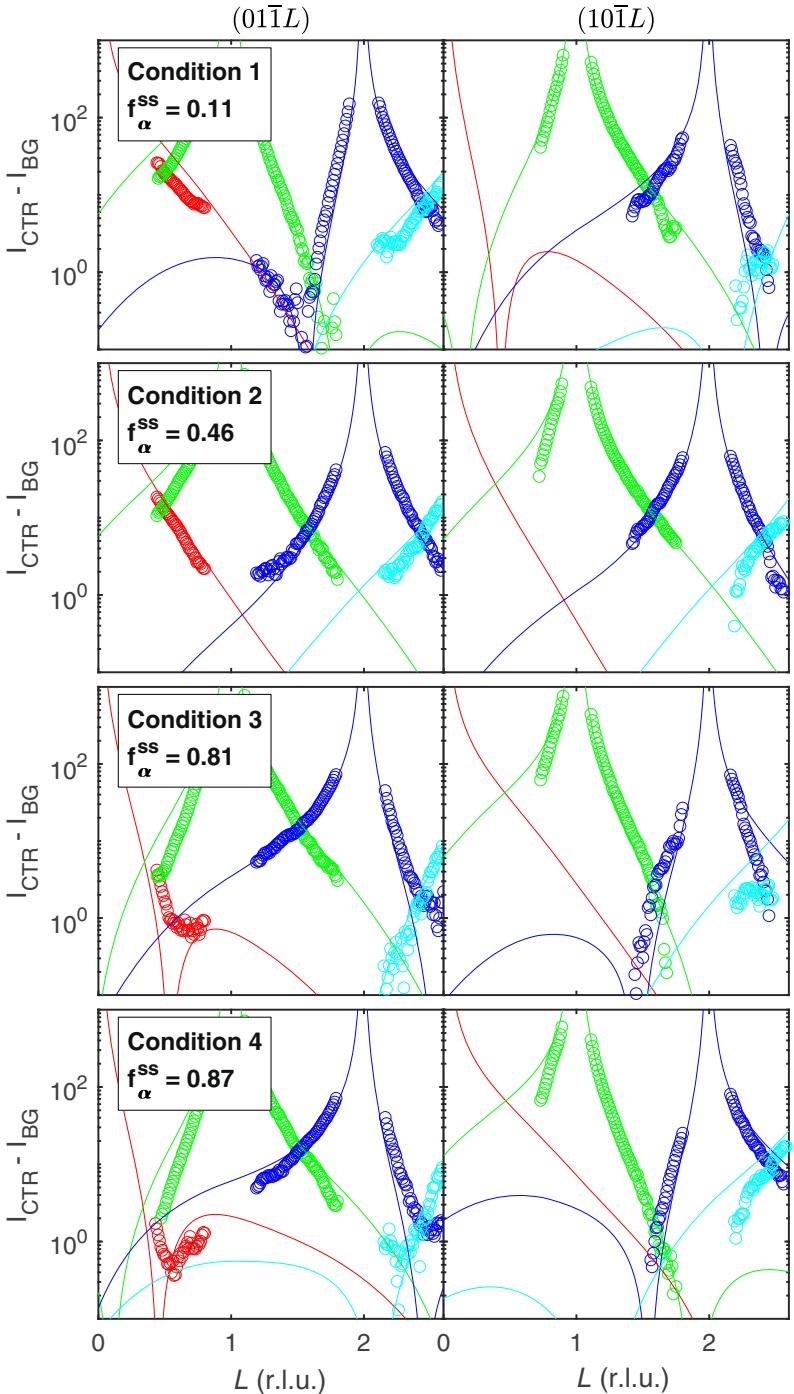

**Fig. 5 Measured and calculated crystal truncation rods.** Symbols show measured net intensities of the ($01\bar{1}L_0$) CTRs and the ($10\bar{1}L_0$) CTRs families (left and right) for CTRs from the $L_0 = 0$, 1, 2, and 3 Bragg peaks at each of four conditions. Curves show fits of all CTRs to obtain steady state $\alpha$ terrace fraction $f_\alpha^{ss}$ at each condition.

To solve this model, we develop a quasi-steady state expression for the dynamics of the terrace fraction $f_\alpha$. Under fairly general assumptions[34], the behavior of $f_\alpha$ can be written as a function of the net growth rate,

$$G = \frac{F - \rho_{eq}^0/\tau}{\rho_0}. \qquad (8)$$

This is simply the difference between the deposition $F$ and a uniform evaporation $\rho_{eq}^0/\tau$, converted to monolayer per second

(ML s$^{-1}$) using the site density $\rho_0$ per half-unit-cell-thick ML. The rate of change in $f_\alpha$ is

$$\frac{df_\alpha}{dt} = K^{dyn}(f_\alpha)\left(\frac{G}{K^{ss}(f_\alpha)} - \frac{4M(f_\alpha)\rho_{eq}^0}{w\rho_0}\right), \qquad (9)$$

where we have introduced the net steady state and dynamic kinetic coefficient functions $K^{ss}(f_\alpha)$ and $K^{dyn}(f_\alpha)$, which in the general case depend on all six $\kappa_x^j$ coefficients[34].

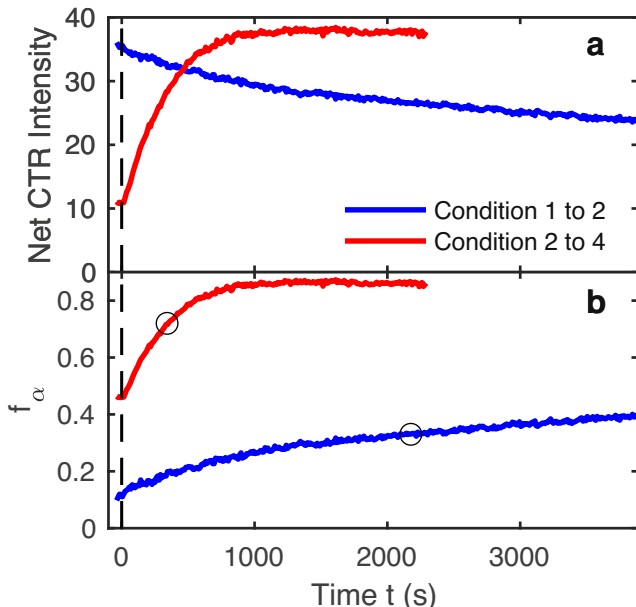

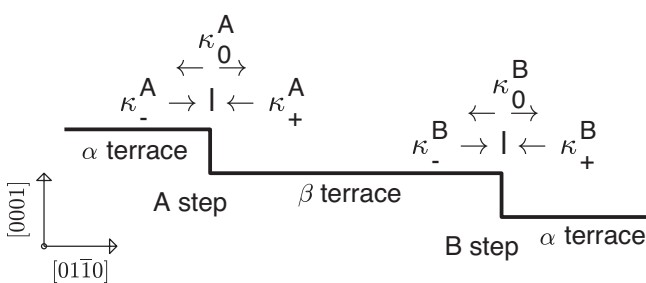

**Fig. 6 Dynamics following a change of condition at t = 0. a** Measured CTR intensities. **b** Calculated terrace fractions $f_\alpha$. Blue curves: conditions 1–2 on the $(10\bar{1}2)$ CTR at $L = 1.627$. Red curves: conditions 2–4 on the $(01\bar{1}2)$ CTR at $L = 1.603$. Circles show $1/e$ relaxation times $t_{rel}$ of 2200 ± 200 and 340 ± 30 s, respectively.

**Fig. 7 Schematic of alternating terraces and steps for the BCF model.** Vicinal {0001} surfaces of HCP crystals have alternating α and β terraces separated by A and B steps. Notations are indicated for the kinetic coefficients for adatom attachment from below and above and for adatom transmission. Schematic of alternating terraces and steps of BCF model for HCP basal-plane surfaces, showing kinetic coefficients for the A and B steps.

The full steady state $df_\alpha/dt = 0$ is obtained at a growth rate of

$$G^{ss}(f_\alpha) = \frac{4\, K^{ss}(f_\alpha)\, M(f_\alpha)\rho_{eq}^0}{w\rho_0}. \qquad (10)$$

This equation for $G^{ss}(f_\alpha)$ can be inverted to obtain the steady state value $f_\alpha^{ss}$ as a function of $G$. The sign of $dG^{ss}/df_\alpha$ and thus $df_\alpha^{ss}/dG$ is determined by the sign of $K^{ss}$. General expressions for $K^{ss}$ show that in most cases, such as that found here, its sign is positive when the kinetic coefficients for $A$ steps are larger than those for $B$ steps[34].

To calculate BCF model results to compare with the experimental conditions, we make three assumptions: (1) the only parameter affected by the TEGa supply rate is the deposition flux $F$; (2) the only parameter affected by the carrier gas composition (0 or 50% $H_2$) is the adatom lifetime $\tau$; and (3) $F$ and $\tau$ enter only through the net growth rate $G$ given by Eq. (8), listed in Table 1 for each condition. We use the known values $\rho_0 = 2a^{-2}/\sqrt{3} = 1.13 \times 10^{19}\ m^{-2}$ and $w = c/\sin(0.52°) = 5.73 \times 10^{-8}\ m$, where $a = 3.20 \times 10^{-10}\ m$ and

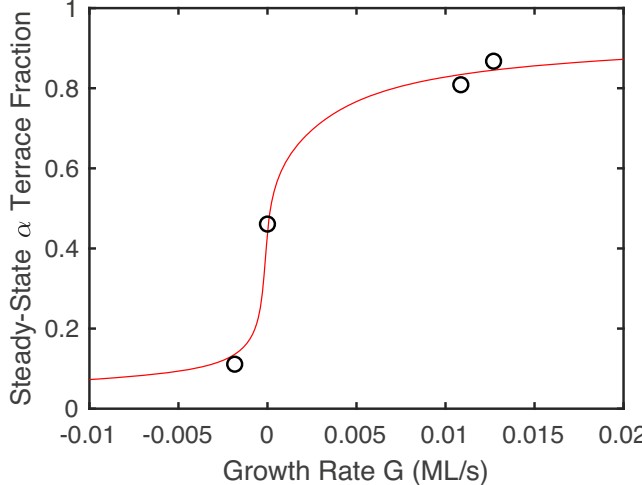

**Fig. 8 Steady state α terrace fraction $f_\alpha^{ss}$ as a function of growth rate G.** Circles are experimental values given in Table 1, showing monotonic increase of $f_\alpha^{ss}$ with increasing $G$. Curve is best-fit BCF model calculation; also fit simultaneously with these four points were the two relaxation times $t_{rel}$ given in Table 1.

$c = 5.20 \times 10^{-10}\ m$ are the lattice parameters of GaN at the growth temperature[42]. We performed fits to the measured quantities (four $f_\alpha^{ss}$ and two $t_{rel}$) using the general expressions for $K^{ss}(f_\alpha)$ and $K^{dyn}(f_\alpha)$[34]. The relaxation times for the model were calculated by integrating Eq. (9) to obtain $f_\alpha(t)$, and then extracting the relaxation time with the same normalization procedure used for the experimental data. The best fit, shown in Fig. 8, was obtained in the limit in which the $\kappa_+^A$ coefficient is large, while the $\kappa_-^A$, $\kappa_-^B$, and $\kappa_0^A$ coefficients approach zero. In this case, $K^{ss}(f_\alpha)$ and $K^{dyn}(f_\alpha)$ can be written as[34]

$$K^{ss}(f_\alpha) = \left[\frac{1}{\kappa_+^B} + \frac{(1 - 2f_\alpha)}{\kappa_0^B} - \frac{wf_\alpha(1 - f_\alpha)}{D}\right]^{-1}, \qquad (11)$$

$$K^{dyn}(f_\alpha) = \left[\frac{1}{\kappa_+^B} + \frac{1}{\kappa_0^B} + \frac{w(1 - f_\alpha)}{D}\right]^{-1}. \qquad (12)$$

Note that $\kappa_+^A$ does not appear in these expressions because in this limit other microscopic processes are rate-limiting. We can fit the measurements directly using these expressions to obtain the four parameters $D/\kappa_+^B = 1.9 \times 10^{-8}\ m$, $D/\kappa_0^B = 1.1 \times 10^{-8}\ m$, $D\rho_{eq}^0\ell^3 = 3.3 \times 10^{-23}\ m^3\ s^{-1}$, and $f_\alpha^0 = 0.44$. Table 1 compares the six measured quantities (four $f_\alpha^{ss}$ and two $t_{rel}$) to the best-fit values calculated from the BCF model.

## Discussion

To interpret the combined parameters obtained from the fits, it is useful to estimate the adatom diffusivity $D$ and equilibrium adatom density $\rho_{eq}^0$. Ab initio calculations of the activation energy for Ga diffusion on the Ga-terminated (0001) surface have given values of $\Delta H_m = 0.4$[43] and 0.5 eV[44], and similar values have been obtained for $3d$ transition metal adatoms[45]. If we estimate the diffusivity from the ab initio calculations using $D = a^2\nu\exp(\Delta S_m/k)\exp(-\Delta H_m/kT)$[46], with $a = 3.2 \times 10^{-10}\ m$, $\nu = 10^{13}\ s^{-1}$, $\Delta S_m = 0$, and $\Delta H_m = 0.4$ eV, we obtain $D = 1.4 \times 10^{-8}\ m^2\ s^{-1}$ at $T = 1073$ K. An analysis of spatial correlations in surface morphology of GaN films[47] indicated a cross-over at $T = 1073$ K from surface diffusion transport to evaporation/condensation transport at a length scale of $\lambda = 1.5 \times 10^{-6}\ m$ for OMVPE growth with $H_2$ present in the carrier gas. Thus, the adatom

lifetime $\tau$ can be estimated as $\tau = \lambda^2/D = 1.7 \times 10^{-4}$ s under these conditions. Using our observed negative net growth rate for $F = 0$ of $G = -\rho_{eq}^0/(\rho_0 \tau) = -0.00184$ ML s$^{-1}$, this gives a value for the equilibrium adatom density of $\rho_{eq}^0 = 3.4 \times 10^{12}$ m$^{-2}$. Using these estimates for $D$ and $\rho_{eq}^0$, the parameters obtained from the mixed kinetics fit imply kinetic coefficients of $\kappa_+^B = 0.74$ m s$^{-1}$ and $\kappa_0^B = 1.3$ m s$^{-1}$, and a step repulsion length of $\ell = 9 \times 10^{-10}$ m.

Although it has not been possible to use scanning-probe microscopy to observe the orientation difference of $\alpha$ and $\beta$ terraces on vicinal basal-plane surfaces of HCP-type systems, our results show that this difference is robustly revealed by surface X-ray scattering. In situ X-ray measurements during growth can determine the fraction covered by each terrace, and thus distinguish the dynamics of $A$ and $B$ steps. While the CTR calculations presented here are for wurtzite-structure GaN, this method applies to many other HCP-type systems with a $6_3$ screw axis, including other compound semiconductors, as well as one-third of the crystalline elements and many more complex crystals.

The BCF model presented here makes predictions for the behavior of the $\alpha$ terrace fraction $f_\alpha$ at steady state and during transients, in terms of surface properties such as the adatom diffusivity $D$ and step kinetic coefficients $\kappa_x^j$. In particular, the steady state fraction $f_\alpha^{ss}$ is predicted to depend only on the net growth rate $G = (F - \rho_{eq}^0 \tau)/\rho_0$, rather than individually on the deposition rate $F$ or the adatom lifetime $\tau$. The positive or negative slope of $f_\alpha^{ss}(G)$ is determined by the sign of a combined kinetic parameter $K^{ss}$.

Our primary experimental result, the positive slope of $f_\alpha^{ss}(G)$, determines the basic nature of the adatom attachment kinetics at $A$ and $B$ steps for GaN (0001) OMVPE. In general, this positive slope implies that $A$ steps have faster kinetics than $B$ steps, that is, the attachment coefficients $\kappa_x^A$ are larger than the $\kappa_x^B$. While a similar general shape of $f_\alpha^{ss}(G)$ is produced by many combinations of the parameters in the BCF model that have faster $A$ than $B$ step kinetics[34], the best fit to our measurements is obtained in the specific limit of Eqs. (11) and (12)). In this limit, the $A$ step has much faster attachment kinetics than the $B$ step, with $\kappa_+^A \gg \kappa_+^B$. This limit also indicates that both $A$ and $B$ steps have standard positive ES barriers, with adatom attachment from below significantly faster than from above for the same supersaturation, and that the $A$ step is nontransparent. We find that $f_\alpha^0$ differs only slightly from the symmetrical value of 1/2.

In comparing the observed values of the step kinetic coefficients to predictions, the 5° rotation of the step azimuth away from [0110] is potentially important, since it determines the minimum average kink spacing on the steps to be 33 Å[1], as discussed in Supplementary Discussion 2. We expect that this relatively small kink spacing will tend to produce higher predicted values of the attachment coefficients $\kappa_+^j$ and $\kappa_-^j$ and lower values of the transmission coefficients $\kappa_0^j$, since attachment occurs when adatoms at a step diffuse along it to find a kink, while transmission occurs when the adatom detaches from the step onto the opposite terrace before it finds a kink[48]. Because the transmitted adatom must traverse the barriers on both sides of the step independent of whether it arrives from above or below, $\kappa_0^j$ does not depend on the direction.

Our conclusion that $A$ steps on GaN (0001) have higher attachment coefficients than $B$ steps agrees with the original qualitative prediction[8], and motivates further quantitative theory development beyond that carried out to date. In several previous studies[6,16–22], various assumptions can lead to the opposite conclusion, as described below. Both the predicted and observed step behavior can depend upon the chemical environment (e.g., OMVPE vs. MBE) and how it passivates the step edges. For example, arguments regarding dangling bonds at steps[6,8] depend

on the effects of very high or low V/III ratios[14] and the presence of NH$_3$ or H$_2$. Likewise, KMC studies[16–19] typically make assumptions about bonding that determine the rates of atomic-scale processes at steps. Detailed ab initio predictions of kinetic barriers and adsorption energies at steps under MBE and OMVPE conditions[20–22] show that they depend strongly on the chemical environment and resulting surface reconstruction. To date, these calculations have been carried out for the Ga(T4), NH(H3) + H(T1), and NH(H3) + NH$_2$(T1) reconstructions. In future theoretical work, it would be useful to consider the specific step-edge structures associated with the 3H(T1) reconstruction found here and in recent calculations for the OMVPE environment[49]. Having an accurate model for atomic incorporation at steps can have important practical implications for advanced GaN devices, for example, laser diodes and white LEDs, by allowing control of interface morphology and incorporation of alloying elements such as In[6,7].

We have demonstrated this method using a micron-scale X-ray beam to satisfy the requirement that the illuminated surface region has a well-defined step azimuth. With current synchrotron X-ray sources, it is convenient to increase the signal rate using a wide-energy-bandwidth pink beam. The higher brightness synchrotron sources soon to come online worldwide will make it possible to perform this type of experiment with highly monochromatic beams, greatly increasing the in-plane resolution of the CTR measurements.

## Methods

**Calculated CTR intensities**. Derivation of the CTR calculations for miscut surfaces with alternating terrace terminations is provided in a separate paper[32]. As described in Supplementary Discussion 1, in fits shown in Table 2, and in recent predictions[49], we expect that the GaN surface under OMVPE conditions has a 3H(T1) reconstruction, in which 3 of every 4 Ga atoms in top-layer sites shown in Fig. 2 are bonded to an adsorbed hydrogen. The calculations in Fig. 3a include the effect of this reconstruction, with equal fractions of all reconstruction domains on both terraces. Since no fractional-order diffraction peaks from long-range-ordered reconstructions are observed in experiments, we expect that the domain structure

**Table 2 Values of fit parameters for each of four OMVPE conditions.**

| Growth condition index | Reconstruction | $f_\alpha^{ss}$ | $\sigma_R$ | $\chi^2$ |
|---|---|---|---|---|
| 1 | 3H(T1) | 0.111 | 0.75 | 106 |
| | Ga(T4) | 0.144 | 1.26 | 130 |
| | NH(H3) + H(T1) | 0.098 | 0.94 | 187 |
| | NH(H3) + NH$_2$(T1) | 0.106 | 0.88 | 200 |
| | NH(H3) | 0.095 | 0.91 | 167 |
| 2 | 3H(T1) | 0.461 | 0.93 | 57 |
| | Ga(T4) | 0.476 | 1.26 | 81 |
| | NH(H3) + H(T1) | 0.460 | 1.15 | 76 |
| | NH(H3) + NH$_2$(T1) | 0.460 | 1.11 | 67 |
| | NH(H3) | 0.459 | 1.13 | 99 |
| 3 | 3H(T1) | 0.811 | 0.85 | 118 |
| | Ga(T4) | 0.670 | 1.46 | 218 |
| | NH(H3) + H(T1) | 0.876 | 1.19 | 205 |
| | NH(H3) + NH$_2$(T1) | 0.869 | 1.15 | 248 |
| | NH(H3) | 0.869 | 1.16 | 168 |
| 4 | 3H(T1) | 0.868 | 0.47 | 80 |
| | Ga(T4) | 0.942 | 1.06 | 112 |
| | NH(H3) + H(T1) | 0.892 | 0.90 | 174 |
| | NH(H3) + NH$_2$(T1) | 0.879 | 0.85 | 220 |
| | NH(H3) | 0.891 | 0.87 | 135 |

We list the values of steady state terrace fraction $f_\alpha^{ss}$, surface roughness $\sigma_R$, and the goodness-of-fit parameter $\chi^2$ from fits to reflectivity for each of five surface reconstructions.

has a short correlation length and all domains are present. We use a surface roughness of $\sigma_R = 0.74$ Å to match the experimental fits described below.

**In situ microbeam X-ray experiments.** We performed in situ measurements of the CTRs in the OMVPE environment at the Advanced Photon Source beamline 12ID-D[50]. At an incidence angle of 2°, the 10 µm X-ray beam illuminated an area of $10 \times 300$ µm. To obtain sufficient signal, we used a wide-bandwidth pink beam setup[51,52]. Details of the measurements and data analysis are given in Supplementary Methods 1. Two types of measurements were performed. We determined the steady state terrace fractions $f_\alpha^{ss}$ under four different growth/evaporation conditions by scanning the detector along the $(01\bar{1}L)$ and $(10\bar{1}L)$ CTRs while continuously maintaining steady state growth or evaporation. We also observed the dynamics of $f_\alpha$ after an abrupt change in conditions.

Under the conditions studied, deposition is transport limited, with the deposition rate proportional to the supply of the Ga precursor (triethylgallium, TEGa), with a large excess of the N precursor ($NH_3$) constantly supplied. We investigated conditions of zero deposition (no supply of TEGa) as well as deposition at a TEGa supply of 0.033 µmol min$^{-1}$. The $NH_3$ flow in both cases was 2.7 s.l.p.m. or 0.12 mol min$^{-1}$, and the total pressure was 267 mbar. The V/III ratio during deposition was thus $3.6 \times 10^6$. For both of these conditions, we studied two carrier gas compositions: 50% $H_2$ + 50% $N_2$ and 0% $H_2$ + 100% $N_2$. The addition of $H_2$ to the carrier gas enhances evaporation of GaN, so that the net growth rate (deposition rate minus evaporation rate) is slightly lower; at zero deposition rate, the net growth rate is negative. We determined the net growth rate for all four conditions as described in Supplementary Methods 2. These values are given in Table 1. Substrate temperatures were calibrated to within ±5 K using laser interferometry from a standard sapphire substrate[50]. While we used the same heater temperature for all conditions, the calibration indicates that the substrate temperature was slightly higher in 50% $H_2$ (1080 K) than in 0% $H_2$ (1073 K). Based on an expected activation energy of ~ 0.4 eV for surface kinetics, this temperature difference produces a negligible (few percent) change in kinetic coefficients.

**Fits to steady state CTRs.** For each growth condition, fits were performed of the calculated CTR intensities to the measured profiles of the $(01\bar{1}L)$ and $(10\bar{1}L)$ CTRs. In addition to a single value of $f_\alpha$, parameters varied in the fit included a single surface roughness $\sigma_R$ and two intensity scale factors, one for each CTR. We fit to log (I) with equal weighting of all points. Fits were done using different possible surface reconstructions. Supplementary Fig. 10 shows the calculated reconstruction phase diagram for the GaN (0001) surface in the OMVPE environment[28], as a function of Ga and $NH_3$ chemical potentials. Based on the chemical potential values that correspond to our experimental conditions estimated in Supplementary Discussion 1, shown by the green rectangle, we considered the five reconstructions highlighted in Supplementary Fig. 10. (The estimate for $\Delta\mu_{Ga}$ has a large uncertainty because it depends on the nitrogen potential produced by decomposition of $NH_3$.) Table 2 shows the values of steady state terrace fraction $f_\alpha^{ss}$, surface roughness $\sigma_R$, and the goodness-of-fit parameter $\chi^2$ from fits to reflectivity at four conditions for each of five reconstructions. The qualitative results for the variation of $f_\alpha^{ss}$ with growth condition are independent of which reconstruction is assumed: $f_\alpha^{ss}$ increases monotonically as the net growth rate $G$ increases. The 3H(T1) reconstruction gives the best fit (minimum $\chi^2$) of the five potential reconstructions, for all four conditions. This is consistent with recent results on GaN (0001) reconstructions in the OMVPE environment[49], which found a phase stability region for the 3H(T1) structure even larger than that shown in Supplementary Fig. 10. Figure 5 compares the fits with the 3H(T1) reconstruction to the measured CTR intensities.

**Extracting terrace fraction dynamics.** We first convert the intensity evolution near $L = 1.6$ to reflectivity $R(t)$ by normalizing it to match the predicted change in reflectivity for the transition in $f_\alpha^{ss}$. We then invert the $R(f_\alpha)$ relation calculated for the experimental $L$ value, shown in Fig. 3b, to obtain $f_\alpha(t)$. For simplicity, we assume the surface roughness is not a function of growth condition, and use the average value of $\sigma_R = 0.74$ Å from the 3H(T1) fits to calculate $R(f_\alpha)$. If the surface roughness were to vary with condition by the amounts shown in Table 2, this would explain only a small fraction of the observed changes (decrease of 7% out of 40% for the transition from conditions 1 to 2, and increase of 16% out of 350% for the transition from conditions 2 to 4). The resulting $f_\alpha(t)$ are shown in Fig. 6b. To extract a characteristic relaxation time for each transition, we interpolate the normalized change $[f_\alpha(t) - f_\alpha(\infty)]/[f_\alpha(0) - f_\alpha(\infty)]$ to obtain the time at which it equals $1/e$.

**Fits of BCF theory to experimental results.** The general expressions for $K^{ss}(f_\alpha)$ and $K^{dyn}(f_\alpha)$[34] involve nine unknown quantities ($D$, $\rho_{eq}^0 \ell^3$, $f_\alpha^0$, and the six $\kappa_x^j$) to be determined or constrained by the measurements. This is a challenge because there are only six measured quantities (four steady state $\alpha$ terrace fractions $f_\alpha^{ss}$ at different growth rates $G$, and two relaxation times for transitions in $G$.) The best fit allowing all nine parameters to vary was first determined by a numerical Levenberg–Marquardt nonlinear regression search algorithm. This minimized the

goodness-of-fit parameter $\chi^2 \equiv \sum [(y_i - y_i^{calc})/\sigma_i]^2$, where the $y_i$ and $\sigma_i$ are the six measured quantities and their uncertainties. To estimate the uncertainties in the $f_\alpha^{ss}$, we multiplied those obtained in the fits to the 3H(T1) reconstruction by a factor of 4, to account for the uncertainties in the atomic coordinates used. We estimated the uncertainty in the $t_{rel}$ to be 10%.

This initial fit found that the minimum $\chi^2$ occurs in the region of parameter space in which the $\kappa_+^A$ coefficient is large, while the $\kappa_-^A$, $\kappa_-^B$, and $\kappa_0^A$ coefficients approach zero. In this case, the limiting expressions (Eqs. (11) and (12)) for $K^{ss}(f_\alpha)$ and $K^{dyn}(f_\alpha)$ involve only the four unknown quantities $D/\kappa_+^A$, $D/\kappa_0^A$, $D\rho_{eq}^0 \ell^3$, and $f_\alpha^0$. A final fit using Eqs. (11) and (12)) allowing only these four parameters to vary gave the same solution as the nine parameter fit.

## Data availability

Data supporting the findings of this study are available within the article and its Supplementary Information file and are available in electronic form from the corresponding author on reasonable request.

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

## Acknowledgements

This work was supported by the US Department of Energy (DOE), Office of Science, Office of Basic Energy Sciences, Materials Science and Engineering Division. Experiments were performed at the Advanced Photon Source beamline 12ID-D, a DOE Office of Science user facility operated by Argonne National Laboratory.

## Author contributions

All authors contributed to initial discussions motivating the experiments and analysis. G. J., C.T., M.J.H., J.A.E., and G.B.S. developed the microbeam surface X-ray scattering method and carried out the measurements. W.W. provided calculated atomic coordinates for the various surface reconstructions. D.X., C.T., P.Z., and G.B.S. developed the CTR and BCF expressions used in the analysis. G.J. and G.B.S. analyzed the results. All coauthors contributed to drafting and editing the manuscript.

## Competing interests

The authors declare no competing interests.
