## [Peer Review File · Nature Communications]

REVIEWER COMMENTS

Reviewer #1 (Remarks to the Author):

Authors report on a very interesting experimental attempt to quantitatively assess the contribution from twin steps, so called A and B atomic step edges, in the growth and decomposition processes of GaN. The topic is very intriguing and, as mentioned by Authors, was previously discussed by many groups. On the other hand, all previous approaches were either theoretical/numerical or were based on ex-situ study that lack the possibility to measure step dynamics and are affected by processes happening during substrate cooling etc.

I have two questions concerning the presented study:

- 1) Why only single GaN substrate was used? This is quite surprising since by using separate substrates in situ XRD results could be compared with AFM or STM measurements. Some assessment of the roughness etc could also be made. Also the effect of parameters such as miscut angle (total value and direction of the miscut angle) on the growth dynamics could be easily addressed.
- 2) Another question related to the previous one is that Authors state that miscut angle direction allows to calculate the density of atomic kinks. But according to BCF theory ([1] page 303-304) main parameter governing kink density is growth temperature. Could Authors comment on why in current manuscript they postulate that kink density can be calculated strictly using miscut angle and how they expect kink density would affect obtained results?

My overall opinion about the paper is high. It is very informative and the presented work is of great value to the field of crystal growth, especially III-nitrides but also other hcp crystals. In spite of this opinion I would like to attract Authors' attention to a few points that could ease the understanding of presented reasoning.

- 1) In Fig 1., where calculated reflectivities for CTR peaks are shown, it might be good to add etiquettes "(01-11)" and "(01-12)" to curves or indicate this in the figure's caption. Also the use of green for terrace beta and reflectivity for (01-11) might be misleading – consider using some darker color for this curve.
- 2) In Fig 2 Authors present the differences between alpha and beta terraces. Authors could consider adding 3D picture of atoms arrangement at A and B step edges to avoid misinterpretation of "A" and "B" edge used in this paper comparing to previous once.
- 3) In the paper, at first, it is hard to follow what "L" and "L_0" stands for. The fact that "L" in figure 3 axis label is non-italic was also misleading to me. I think that some single-sentence explanation just after the first use in section Results will help greatly.
- 4) On page 4. Authors state: "A KMC study of growth on an HCP lattice¹⁹ found a much lower Ehrlich-Schwoebel (ES) barrier at B steps than at A steps...". Of course it should be "ab initio" instead of "KMC".

Regards,
Henryk Turski

Reviewer #2 (Remarks to the Author):

This is an overall well-written manuscript on the use of X-ray microbeams to investigate the kinetics of alternating A and B steps on vicinal {0001} HCP surfaces during MOCVD growth. The work is surprisingly quantitative, using an extension of the classic BCF theory to obtain kinetic coefficients for A and B steps. The overall conclusion that the attachment rate constants are higher for A steps seems overall well supported, aside from some minor concerns, which I will describe below. The manuscript is suitable for publication with minor revisions.

Minor items that could be clarified or improved appear below.

One concern is that the manuscript is not self-contained since the growth model is only outlined, and Ref 36 is still a preprint.

Some questions related to the model:

1) Relaxation time t_{rel} -- Experimental values are described clearly, but it is not clearly explained how this quantity is obtained from the model. Can this be described instead of just referring to Ref 36? There appears to be a typo in the paragraph just below Eq 12 (t_{ref} instead of t_{rel}).

2) k_{+A} is large but doesn't appear in equations 11 and 12. Does it drop out because it is assumed to be large?

3) What is the thinking behind κ_{0j} the kinetic coefficient for transmission across a step? Is this process distinct from attachment from above, followed by detachment to the lower terrace, i.e. a two-step process? Some discussion would be appreciated. Fig. 7 seems to indicate that the coefficient is the same for adatoms impinging from above or below, which is counterintuitive.

4) P12: "...in which various assumptions can lead to the opposite conclusion." Please clarify what assumptions lead to opposite conclusions. Do the authors intended this to be an expression of uncertainty about the model presented in the manuscript?

A question on Fig 2c: Do A & B steps ever merge into double height steps? Fig 2c seems to show steps merging in some places. But perhaps this is due to limited spatial resolution in the AFM scan.

Methods: What is the thickness of the glass that makes up the wall of the MOCVD reactor? Is there significant background from the glass? What is the percentage of the x-rays are absorbed?

Response to Reviewers' Comments (NCOMMS-20-45729)

We'd like to thank the reviewers for their insightful comments, questions, and suggestions for clarification. We are delighted and grateful that the changes made at the suggestion of the reviewers have improved the manuscript. Changes made to the manuscript and its supplemental information are highlighted in green text in the revised versions. Detailed responses to the Reviewer's comments are given here.

(Reviewers' comments in italics)

Reviewer #1 (Remarks to the Author): Authors report on a very interesting experimental attempt to quantitatively assess the contribution from twin steps, so called A and B atomic step edges, in the growth and decomposition processes of GaN. The topic is very intriguing and, as mentioned by Authors, was previously discussed by many groups. On the other hand, all previous approaches were either theoretical/numerical or were based on ex-situ study that lack the possibility to measure step dynamics and are affected by processes happening during substrate cooling etc.

I have two questions concerning the presented study:

- (1) Why only single GaN substrate was used? This is quite surprising since by using separate substrates in situ XRD results could be compared with AFM or STM measurements. Some assessment of the roughness etc could also be made. Also, the effect of parameters such as miscut angle (total value and direction of the miscut angle) on the growth dynamics could be easily addressed.*

Response: As the reviewer notes above, an advantage of the *in-situ* X-ray measurement of the terrace fraction is that it avoids questions of whether the terrace fraction changed during cooling and removal for *ex-situ* study. While the agreement of the terrace fraction observed *ex situ* by AFM (Fig 4) and the corresponding *in-situ* X-ray measurement for the zero-growth-rate condition is gratifying, probably only such zero-growth-rate conditions could be 'quenched' in our system for *ex-situ* study, based on the dynamics we observed (e.g. Fig 6). Other structures would evolve in an undetermined way during the hour-long cooling of the sample at zero growth rate required to extract it from the chamber.

Our X-ray method also unambiguously differentiates whether the alpha or beta terrace fraction is larger, unlike AFM which cannot distinguish between alpha and beta.

The advantage of using a single sample for the comparison of different growth rates and HK positions is that it avoids the question of whether the sample miscut amplitude and direction may also differ. We felt it was critical to take CTR scans at different HK positions for a grid of different growth conditions using the same sample, so that any differences could be attributed to these variables alone.

Furthermore, synchrotron beamtime for these measurements was limited, since this was only part of a study that also included XPCS measurements during step-flow growth. It

takes about a full work day for the process of cooling a sample to room temperature, installing a new sample, aligning it, and characterizing its miscut and growth behavior at high temperature, before taking CTR data on terrace widths vs conditions. The primary variables to which this would give access, e.g. miscut amplitude and azimuth, while potentially interesting, are likely secondary to the effects of growth conditions.

We hope that in the future we will be able to revisit this system with additional beamtime to carry out measurements at a larger number of growth rates, temperatures, and miscuts, in order to further quantify the kinetic parameters and their dependences. But the current results already clearly determine the fundamental difference between the growth behavior of A and B steps.

- (2) *Another question related to the previous one is that Authors state that miscut angle direction allows to calculate the density of atomic kinks. But according to BCF theory ([1] page 303-304) main parameter governing kink density is growth temperature. Could Authors comment on why in current manuscript they postulate that kink density can be calculated strictly using miscut angle and how they expect kink density would affect obtained results?*

Response: You are correct to point out our oversight of the contribution of thermally generated kinks. The miscut angle direction only places a lower limit on the density of kinks; this density can be higher if additional kink pairs are generated thermally. While we can easily calculate this geometrical minimum kink density (or maximum kink spacing) due to the miscut direction, we must estimate the contribution of thermally generated kinks by assuming a formation energy for kinks. The geometrical maximum spacing of kinks for a step at the measured 5 degree rotation is a/θ , where the difference in positive and negative kink probability is $\theta = n_+ - n_- = 2/[\sqrt{3}/\tan(5 \text{ deg}) + 1]$ for the close-packed lattice. This spacing is $10.4 a$, where a is the lattice parameter. Using the analysis in [1], with the energy to create a kink pair estimated as $2w = W/6$ where $W = 3.38 \text{ eV}$ is the bulk binding energy per molecule for GaN [19], gives a thermally generated kink spacing for a crystallographically aligned step of $[\exp(w/kT) + 2] a/2 = 11.6 a$. The combination of thermally generated and geometrical kinks for a 5 degree rotation is given by $a/(\theta + 2n_-)$, where n_- satisfies $(\theta + n_-)n_- = \exp(-2w/kT)$. This gives a slightly smaller spacing of $7.4 a$.

A different kink density does not affect the experimental results (i.e. the observed difference in the growth behavior of A and B steps, and the fit values of the kinetic coefficients). As mentioned in the Discussion section, it could affect the comparison of the fit values to predicted values, since a large increase in kink density would be expected to make step transparency smaller, i.e. explain lower values of κ_j^i . Given the large uncertainty in the thermally generated kink density, the well-defined minimum density given by the miscut direction helps make the interpretation of the observed values less uncertain.

We have changed the text on pages 6 and 12 to say that the miscut angle direction determines the *minimum* kink density, have given the maximum kink spacing and mentioned that it could be smaller due to thermally generated kinks, and have added the above analysis to the Supplementary Information as Supplementary Discussion 2.

My overall opinion about the paper is high. It is very informative, and the presented work is of great value to the field of crystal growth, especially III-nitrides but also other hcp crystals. In spite of this opinion I would like to attract Authors' attention to a few points that could ease the understanding of presented reasoning.

- (1) *In Fig 1., where calculated reflectivities for CTR peaks are shown, it might be good to add etiquettes “(01-11)” and “(01-12)” to curves or indicate this in the figure’s caption. Also, the use of green for terrace beta and reflectivity for (01-11) might be misleading – consider using some darker color for this curve.*

Response: Thank you for pointing out the issue in Fig. 1. We have changed the color of the reflectivity curve for (01-11) from green to red, and specified the colors of the (01-11) and (01-12) curves in the caption. We also improved the legibility of the text in Fig. 1, and clarified that the typical values of f_α given for evaporation and deposition apply only to GaN (0001) OMVPE.

- (2) *In Fig 2 Authors present the differences between alpha and beta terraces. Authors could consider adding 3D picture of atoms arrangement at A and B step edges to avoid misinterpretation of “A” and “B” edge used in this paper comparing to previous ones.*

Response: We agree that additional definition of the A and B steps is desirable since they are central to the paper. We have added a cross-section of the step structure into Fig. 2 to clearly differentiate the A and B step structures.

- (3) *In the paper, at first, it is hard to follow what “L” and “L_0” stands for. The fact that “L” in figure 3 axis label is non-italic was also misleading to me. I think that some single-sentence explanation just after the first use in section Results will help greatly.*

Response: We have added a single-sentence explanation in section Results on page 5 to clarify the meaning of L_0 : “Bragg peak locations have integer indices $H_0K_0L_0$ in reciprocal lattice units; these indices identify the CTR associated with each peak.” The axis label with L has been changed from non-italic to italic format in Figs. 1, 3, and 5.

- (4) *On page 4. Authors state: “A KMC study of growth on an HCP lattice [19] found a much lower Ehrlich-Schwoebel (ES) barrier at B steps than at ...”. Of course, it should be “ab initio” instead of “KMC”.*

Response: Reference 19 is actually a KMC study. The different predicted ES barriers for A and B steps result from the standard bond-counting energetics used in the KMC model. We have changed the description of the KMC study on page 4 to clarify this.

Reviewer #2 (Remarks to the Author): This is an overall well-written manuscript on the use of X-ray to investigate the kinetics of alternating A and B steps on {0001} HCP surfaces during MOCVD growth. The work is surprisingly quantitative, using an extension of the classic BCF theory to obtain kinetic coefficients for A and B steps. The overall conclusion that the attachment rate constants are higher for A steps seems overall well supported, aside from some minor concerns, which I will describe below. The manuscript is suitable for publication with minor revisions. Minor items that could be clarified or improved appear below.

One concern is that the manuscript is not self-contained since the growth model is only outlined, and Ref 36 is still a preprint.

Response: Due to the wider-ranging theory results in [36], not all of which are directly relevant to this experimental study, and the space limitations in this paper, it seems best to us to summarize the relevant part of the BCF model for alternating step types in this paper, and to make available the more detailed development of the theory in a preprint. We are finishing up this detailed treatment and plan to have it submitted soon to an archival journal.

Some questions related to the model:

- (1) Relaxation time t_{rel} -- Experimental values are described clearly, but it is not clearly explained how this quantity is obtained from the model. Can this be described instead of just referring to Ref 36? There appears to be a typo in the paragraph just below 12 (t_{ref} instead of t_{rel}).*

Response: Thank you for pointing out this omission. The relaxation times for the model were calculated by integrating Eq (9) to obtain $f_a(t)$, and then extracting the relaxation time with the same normalization procedure used for the experimental data. We have added a sentence describing this to Results page 11 above Eq. (11). The typo in the paragraph just below equation 12 has also been fixed.

- (2) κ_+^A is large but doesn't appear in equations 11 and 12. Does it drop out because it is assumed to be large?*

Response: Yes, in the mixed kinetic limit with large κ_+^A , it does not appear in the expressions (11) and (12) because other processes are rate-limiting. We have added a statement to this effect after Eq. (12).

- (3) What is the thinking behind κ_j^j the kinetic coefficient for transmission across a step? Is this process distinct from attachment from above, followed by detachment to the lower terrace, i.e. a two-step process? Some discussion would be appreciated. Fig. 7 seems to indicate that the coefficient is the same for impinging from above or below, which is counterintuitive.*

Response: The adatom transmission process has been described in the literature, e.g. ref [50]. It is distinct from adatom attachment, which involves an adatom arriving at a step and then diffusing along the step to reach a kink, where it is incorporated into the crystal; adatom transmission involves the adatom detaching from the step onto the opposite

terrace, before it finds a kink. Because the transmitted adatom must traverse the barriers on both sides of the step independent of whether it arrives from above or below, the kinetic coefficient is the same. This is related to the reason that the kink density can affect the relative sizes of the various kinetic coefficients, as discussed in the second question of Reviewer 1. We have added some sentences to explain adatom transmission in Page 12 near ref [50].

- (4) *P12: "...in which various assumptions can lead to the opposite conclusion." Please clarify what assumptions lead to opposite conclusions. Do the authors intend this to be an expression of uncertainty about the model presented in the manuscript?*

Response: Thank you for pointing out this ambiguous wording. This first sentence of this paragraph is meant to introduce the sentences that follow, which discuss in more detail the assumptions used in previous theory and simulation work that can lead to the opposite conclusion. We have split the sentence and reworded it to clarify that the assumptions that lead to the opposite conclusion are in the previous literature and are discussed in the following sentences.

- *A question on Fig 2c: Do A & B steps ever merge into double height steps? Fig 2c seems to show steps merging in some places. But perhaps this is due to limited spatial resolution in the AFM scan.*

Response: Due to the limited spatial resolution of these AFM images, it is hard to be sure whether some step pairs locally merge into double-height steps on an atomic scale or simply are very close together in some places. The BCF theory includes a step repulsion term that keeps the steps from merging, and the average terrace fractions we observe in the X-ray experiments do not approach too closely to 0 or 1, which would correspond to all step pairs merging into double-height steps.

- *Methods: What is the thickness of the glass that makes up the wall of the MOCVD reactor? Is there significant background from the glass? What is the percentage of the x-rays are absorbed?*

Response: While the wall thickness of the fused quartz chamber described in reference [52] is 2 mm, which will absorb about 60% total for 4 mm at the 25.75 keV x-ray energy used, for these experiments we actually employed a modified chamber which had beryllium entrance and exit windows. These absorb a negligible amount. Slits downstream of the exit window block the scattering from the windows from reaching the region of the detector used to collect the CTR signal. We have added sentences describing the Be windows and slits to Supplementary Information page 1, section Supplementary Methods 1: X-ray Scattering.

Unrelated to any reviewer comments, we updated the captions of Figs. S7, S10, and S11 in the Supplementary Information to clarify the presentation, and added a sentence to the Author Contributions section.

REVIEWERS' COMMENTS

Reviewer #1 (Remarks to the Author):

I am fully satisfied with all answers and corrections introduced by Authors. Updated version of the manuscript is clear and easy to follow. I believe it can be published in Nature Communications as is.

Reviewer #2 (Remarks to the Author):

The revised manuscript fully addresses my concerns. I believe it is ready to be published.